American Society for Microbiology | Microbiology Spectrum
# Anti-Arbovirus Antibodies Cross-React With Severe Acute Respiratory Syndrome Coronavirus 2

Taweewun Hunsawong,[a] Darunee Buddhari,[a] Kamonthip Rungrojcharoenkit,[a] Rungarun Suthangkornkul,[a] Duangrat Mongkolsirichaikul,[a] Jindarat Lohachanakul,[a] Kedsara Tayong,[a] Kanittha Sirikajornpan,[a] Prinyada Rodpradit,[a] Yongyuth Poolpanichupatam,[a] Chonticha Klungthong,[a] Darunee Utennam,[b] Surachai Kaewhiran,[c] Thomas S. Cotrone,[a] Stefan Fernandez,[a] Anthony R. Jones[a]

[a]Department of Virology, US Army Medical Directorate, Armed Forces Research Institute of Medical Sciences, Bangkok, Thailand
[b]Research Division, Royal Thai Army, Armed Force Research Institute of Medical Sciences, Bangkok, Thailand
[c]Kamphaeng Phet Hospital, Kamphaeng Phet, Thailand

**ABSTRACT** Severe acute respiratory syndrome coronavirus 2 (SARS-CoV-2) is found in regions where dengue (DENV) and chikungunya (CHIKV) viruses are endemic. Any serological cross-reactivity between DENV, CHIKV, and SARS-CoV-2 is significant as it could lead to misdiagnosis, increased severity, or cross-protection. This study examined the potential cross-reactivity of anti-DENV and CHIKV antibodies with SARS-CoV-2 using acute and convalescent-phase samples collected before the SARS-CoV-2 pandemic. These included healthy, normal human (NHS, $n = 6$), CHIKV-positive ($n = 14$ pairs acute and convalescent), primary DENV-positive ($n = 20$ pairs), secondary DENV-positive ($n = 20$ pairs), and other febrile illnesses sera ($n = 23$ pairs). Samples were tested using an in-house SARS-CoV-2 and a EUROIMMUN IgA and IgG ELISAs. All NHS samples were negative, whereas 3.6% CHIKV, 21.7% primary DENV, 15.7% secondary DENV, and 10.8% febrile diseases sera resulted as anti-SARS-CoV-2 antibody positive. The EUROIMMUN ELISA using spike 1 as the antigen detected more positives among the primary DENV infections than the in-house ELISA using spike 1-receptor binding domain (RBD) protein. Among ELISA-positive samples, four had detectable neutralizing antibodies against SARS-CoV-2 reporter virus particles yet none had detectable neutralizing antibodies against the live Wuhan strain of SARS-CoV-2. These data demonstrated the SARS-CoV-2 diagnostic cross-reactivity, but not neutralizing antibody cross-reactivity, among dengue seropositive cases.

**IMPORTANCE** SARS-CoV-2 continues to cause significant morbidity globally, including in areas where DENV and CHIKV are endemic. Reports using rapid diagnostic and ELISAs have demonstrated that serological cross-reactivity between DENV and SARS-CoV-2 can occur. Furthermore, it has been observed that convalescent DENV patients are at a lower risk of developing COVID-19. This phenomenon can interfere with the accuracy of serological testing and clinical management of both DENV and COVID-19 patients. In this study, the cross-reactivity of primary/secondary anti-DENV, CHIKV, and other febrile illness antibodies with SARS-CoV-2 using two ELISAs has been shown. Among ELISA-positive samples, four had detectable levels of neutralizing antibodies against SARS-CoV-2 reporter virus particles. However, none had detectable neutralizing antibodies against the live Wuhan strain of SARS-CoV-2. These data demonstrated SARS-CoV-2 diagnostic cross-reactivity, but not neutralizing antibody cross-reactivity, among dengue seropositive cases. The data discussed here provide information regarding diagnosis and may help guide appropriate public health interventions.

**KEYWORDS** Chikungunya virus, Dengue virus, SARS-CoV-2, serological cross-reactivity, arbovirus, cross-protection, neutralizing antibodies

Address correspondence to Taweewun Hunsawong, Taweewunh.fsn@afrims.org, or Anthony R. Jones, Anthony.Jones.mil@afrims.org.

The authors declare no conflict of interest.

**D**engue virus (DENV) has been the main cause of arboviral disease in tropical and subtropical areas over numerous decades (1). DENV, of the flavivirus family, has four closely related, but antigenically distinctive, serotypes (DENV-1 to DENV-4). Infection with any of the DENV serotypes will provide long-term protection against a homotypic infection but only short-term protection against heterotypic infections (2). Clinical manifestations can range from flu-like symptoms to more severe symptoms, including dengue hemorrhagic fever (DHF) and dengue shock syndrome (DSS). Geographically overlapping with DENV, Chikungunya virus (CHIKV), an emerging alphavirus of the *Togaviridae* family, is transmitted to humans through the bite of an *Aedes* mosquito (3–6). Clinical manifestations of a CHIKV infection are usually characterized by an abrupt onset of high fever, profound joint pain, rash, and arthralgia.

Severe acute respiratory syndrome coronavirus 2 (SARS-CoV-2), the causative agent of coronavirus disease 2019 (COVID-19), continues to cause significant morbidity globally. More than half of the world's population has been fully vaccinated, but new variants of concern are capable of escaping, at least to some degree, vaccine-derived protection (7, 8). Clinical manifestations of SARS-CoV-2 infections range from mild (fever, cough, tiredness, sore throat, and diarrhea) to severe (difficulty breathing, shortness of breath, and death) (9). Early diagnosis and treatment are important factors to minimize the risk of severe symptoms in patients and curbing new cases. In addition to RT-PCR and antigen testing, IgA and IgG antibody detection are also used to identify SARS-CoV-2 infection and document SARS-CoV-2 seroprevalence.

Serological cross-reactivity between DENV and SARS-CoV-2 has been reported when using lateral flow rapid diagnostic tests and ELISAs (10–13). There is no similar data available for CHIKV. Computational-based studies show amino acid homology between the DENV envelope protein and the SARS-CoV2 spike protein. More specifically, the DENV envelope protein has homologous amino acid sequences with the heptapeptide repeat sequence 2 (HR2) domain in the spike 1 subunit (14). Consequently, a possibility of anti-DENV antibodies binding to SARS-CoV-2 spike 1-RBD exists (15). COVID-19 patients who were previously exposed to DENV have been shown to have a lower risk of death than DENV-naive patients, further substantiating the possibility that dengue infections can induce some level of immunological protection against SARS-CoV-2 (16). If it exists, cross-reactivity between DENV and SARS-CoV-2 would interfere with the accuracy of serological testing and clinical management of patients. This study aimed to demonstrate the potential cross-reactivity and cross-neutralization of DENV and CHIKV antibodies with SARS-CoV-2 using well-characterized sample panels collected before the SARS-CoV-2 pandemic. These pre-COVID-19 serum samples were tested for anti-SARS-CoV-2 binding and neutralizing antibodies using in-house and commercial IgA/IgG ELISAs, a SARS-CoV-2 reporter virus particle, and a live microneutralization assay.

## RESULTS

**Serum panel.** To demonstrate the cross-reactivity between arboviral antibodies and SARS-CoV-2 serological testing, five serum panels were selected. These serum panels included normal human serum (NHS) from blood donors (*n* = 6), febrile illness (*n* = 23), primary DENV infection (*n* = 20), secondary DENV infection (*n* = 20), and CHIKV infection (*n* = 14) (Table 1). Acute and convalescent-phase sera were obtained at 0 to 5 days and 5 to 16 days post-onset, respectively, from all groups except NHS where only one blood collection was available. All samples were collected from January to November 2019. Samples were tested for DENV and CHIKV using reverse transcription (RT)-PCR and an immunoglobulin (Ig)M/IgG enzyme-linked immunosorbent assay (ELISA). NHS and febrile illness samples were confirmed to be negative by both DENV/CHIKV RT-PCR and DENV/JEV/CHIKV IgM/IgG ELISAs. The DENV and CHIKV serum panels were confirmed to be DENV or CHIKV positive, respectively, either by RT-PCR or ELISA. DENV immune status (primary or secondary infection) was determined by the ratio of IgM over IgG levels as described previously (17).

**TABLE 1** Characteristics of serum panels used for the investigation of the cross-reactivity of arboviral antibodies with SARS-CoV-2

| Group | $n$ (acute/convalescent pairs) | PCR | IgM/IgG EIA |
|---|---|---|---|
| Normal human serum (blood donor) | 6 (single blood collection) | Arbovirus negative | Arbovirus negative |
| Febrile illness | 23 | Arbovirus negative | Arbovirus negative |
| Primary dengue infection | 20 | Dengue virus | Primary dengue infection |
| Secondary dengue infection | 20 | Dengue virus | Secondary dengue infection |
| Chikungunya virus infection | 14 | Chikungunya virus | Chikungunya virus |

**SARS-CoV-2 IgA and IgG binding antibodies.** Sample cross-reactivity to SARS-CoV-2 was initially assessed using the EUROIMMUN ELISA with whole spike 1 as antigen and further tested against the more specific target antigen, spike 1-receptor binding domain (RBD), by an in-house ELISA. EUROIMMUN and in-house ELISA testing revealed that no NHS samples had detectable anti-SARS-CoV-2 IgA/IgG antibodies. Conversely, 3.6% ($n$ = 3), 21.7% ($n$ = 18), 15.7% ($n$ = 13), and 10.8% ($n$ = 9) of CHIKV, primary DENV, secondary DENV, and febrile disease samples, respectively, were positive by either of ELISA format. For IgA ELISA, most positive samples fell under the primary DENV infection serum panel (19.3%, $n$ = 16) followed by the secondary DENV infection panel (13.3%, $n$ = 11), the febrile illness panel (3.6%, $n$ = 3), and the CHIKV panel (2.4%, $n$ = 2). EUROIMMUN IgA ELISA detected a significantly higher ($P$ = 0.0001) number of positive samples than the in-house ELISA. This was most evident in the primary DENV panel ($n$ = 14) (Fig. 1A) followed by the secondary DENV panel ($n$ = 5) and the febrile disease panel ($n$ = 2). No CHIKV or NHS samples showed positive results. For the in-house ELISA, a significant ($P$ < 0.0001) increase in IgA optical density (OD) levels was observed in the secondary DENV infections panel compared to other groups (Fig. 1B). The secondary DENV infections serum panel had the most anti-SARS-CoV-2 positive samples ($n$ = 6) followed by the primary DENV panel ($n$ = 2), CHIKV ($n$ = 1), and the febrile illness panel ($n$ = 1). Nevertheless, the proportion of IgA-positive samples using the in-house ELISA was significantly lower ($P$ = 0.010) than the EUROIMMUN ELISA.

For the anti-SARS-CoV-2 IgG ELISAs, most positive samples were also found in the primary DENV infection panel (9.6%, $n$ = 8) followed by the secondary DENV infection panel (3.6%, $n$ = 3) and the febrile illness panel (2.4%, $n$ = 2). No positive samples were detected in the CHIKV or NHS serum panels. EUROIMMUN ELISA (Fig. 1C) detected positive samples in two panels, the primary DENV panel ($n$ = 6) and the secondary DENV infection panel ($n$ = 1). Conversely, the in-house ELISA detected similar proportions of positive samples among the secondary DENV, primary DENV, and febrile illness panels ($n$ = 2 per group) (Fig. 1D). In addition to the serum panels, the cross-reactivity of MAb specific to flavivirus (4G2), DENV group (2H2), and DENV-1 to DENV-4 (1F1, 3H5, 8A1 and 1H10) with SARS-CoV-2 IgA/IgG ELISAs were tested. It was found that none of these monoclonal antibodies (MAbs) had a detectable OD value equal to or higher than the positive cutoff (unpublished data).

Using Pearson's correlation, negative correlations were demonstrated between the in-house SARS-CoV-2 IgA and DENV IgM ($r$ = −0.016, $P$ = 0.8474)/IgG ($r$ = −0.103, $P$ = 0.0366), as well as between EUROIMMUN IgA with DENV IgG ($r$ = −0.103, $P$ = 0.2534) and between EUROIMMUN IgG with DENV IgG ($r$ = −0.092, $P$ = 0.0327). A poor correlation was found between the in-house SARS-CoV-2 IgG and DENV IgM ($r$ = 0.229, $P$ = 0.0045)/IgG ($r$ = 0.055 $P$ = 0.5378). EUROIMMUN IgA ($r$ = 0.382, $P$ < 0.0001) and IgG ($r$ = 0.411, $P$ < 0.0001) showed a moderate correlation with DENV IgM. Anti-SARS-CoV-2 IgA/IgG antibodies detected by the in-house and EUROIMMUN ELISA were not related to days post onset of sample collection (unpublished data).

**Neutralizing antibody against SARS-CoV-2 Wuhan strain.** Anti-SARS-CoV-2 IgA (Fig. 2A, $n$ = 30) and IgG (Fig. 2B, $n$ = 12) positive samples by either EUROIMMUN or in-house ELISAs were tested for functional antibodies by both RVP and live SARS-CoV-2 microneutralization assays for two main reasons. These were (i) to confirm positive

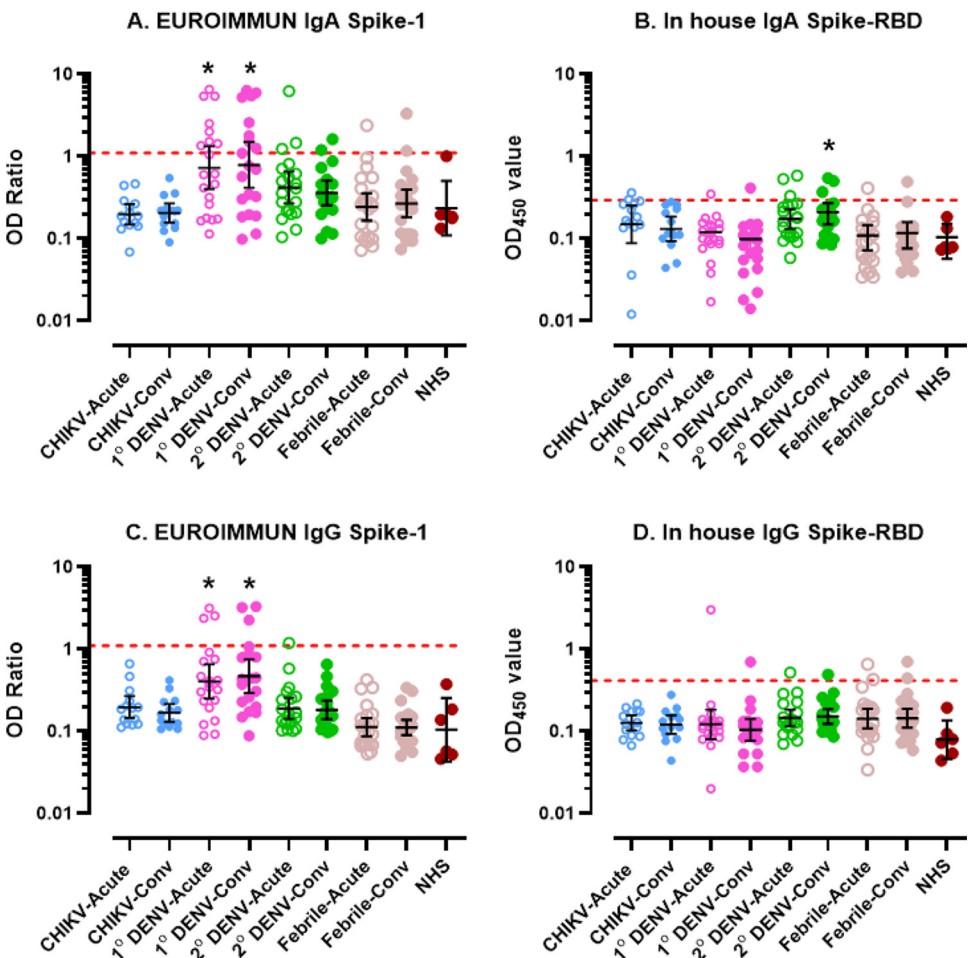

**FIG 1** Anti-SARS-CoV-2 binding antibody detected by ELISA. Anti-SARS-CoV-2 IgA and IgG antibodies in CHIKV, 1° DENV, 2° DENV, febrile illness, and NHS samples were identified by using two ELISAs. EUROIMMUN IgA (A)/IgG (B) using spike 1 as the target antigen demonstrated that most of the positive samples fell under 1° DENV infection ($P < 0.0001$). The in-house IgA (C)/IgG (D) using the spike 1 receptor binding domain (RBD) as antigen showed the majority of positive samples fell under 2° DENV infection. The red dotted line indicates the positive OD cutoff. The red dot line indicates a positive OD Ratio cut off. CHIKV, Chikungunya virus; 1° DENV, primary dengue virus infection; 2° DENV, secondary dengue infection; Conv, convalescent. *, $P < 0.05$.

results and (ii) to investigate whether antibody production during arboviral infection could mediate cross-neutralization of SARS-CoV-2. Neutralizing antibodies against the SARS-CoV-2 Wuhan strain using SARS-CoV-2 RVP microneutralization assay showed four samples from four patients who had detectable $NT_{50}$ antibodies at low levels. Two of them were febrile illness samples ($NT_{50}$ titers at 20 and 47), one from the primary DENV infection group ($NT_{50}$ titer at 15), and one from the secondary DENV infection group ($NT_{50}$ titers at 13). Because the RVP contains only spike 1-RBD glycoprotein, we further determined the levels of neutralizing antibodies against the live SARS-CoV-2 Wuhan strain. From four RVP positives, none had detectable $NT_{50}$ titers.

## DISCUSSION

Soon after COVID-19 was initially identified in Wuhan, China at the end of 2019, the disease rapidly spread worldwide. This included regions with endemic arboviral diseases like DENV and CHIKV. Diagnosis of SARS-CoV-2, DENV, and CHIKV relies on PCR and/or antibody testing. Studies have demonstrated the cross-reactivity of DENV antibodies with SARS-CoV-2 using commercial rapid test or IgA, IgM, IgG ELISAs testing on single serum specimens collected from DENV patients. Recently, a study in Brazil

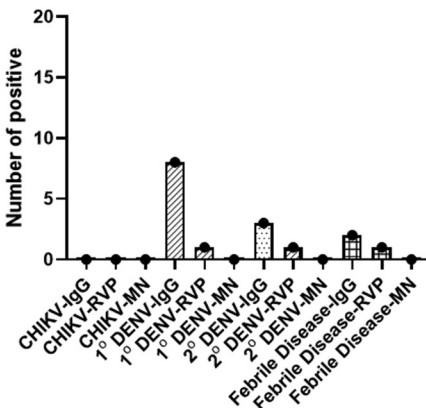

**FIG 2** Number of positive samples in each group detected by binding ELISAs and a microneutralization assay against SARS-CoV-2 reporter virus particles and live virus. The anti-SARS-CoV-2 IgA (A) and IgG (B) positive samples, determined by either in-house or EUROIMMUN ELISAs were tested for neutralizing antibodies against the SARS-CoV-2 Wuhan strain using SARS-CoV-2 reporter virus particles (RVPs) and a live virus microneutralization assay. Several 1° DENV and 2° DENV infections, including febrile diseases samples were positive when tested against RVPs, but none of them had detectable neutralizing antibodies against the live virus. CHIKV, chikungunya virus; 1° DENV, primary dengue virus infection; 2° DENV, secondary dengue infection.

demonstrated that DENV preexposure was associated with lower mortality rates after COVID-19 infection. These findings raise the possibility that dengue might induce a degree of immunological protection against SARS-CoV-2 infection (16). Additionally, COVID-19 transmission from March to April 2020 was lower in DENV areas of endemicity, including Latin America, Africa, SE Asia, and the Indian subcontinent than in regions that have less circulating DENV, like Europe and North America (18).

Serological cross-reactivity between DENV and SARS-CoV-2 virus is important as it can lead to misdiagnosis for both diseases and misunderstandings about the potential for cross-protection. This study used well-characterized serum panels collected before the SARS-CoV-2 pandemic and specific monoclonal antibodies to demonstrate the cross-reactivity and cross-protection of DENV and CHIKV antibodies with SARS-CoV-2 IgA/IgG binding antibody testing using two ELISA formats and microneutralization assays. The results indicated that an in-house ELISA using spike 1-RBD as antigen could detect antibody cross-reactivity at lower levels than those in a commercial ELISA using whole spike 1 as antigen. *In silico* analysis of the DENV envelope protein and the heptapeptide repeat sequence 2 (HR2) domain in spike 1 subunit of SARS-CoV2 cross-reactivity has been reported (14). This supports our findings as the HR2 is located at amino acid residuals 116 to 1213 in spike 1 protein which does not overlap the region of the RBD (residuals 319 to 541) (19). However, both ELISAs were unable to detect any positive results when tested with MAb specific to the conserved region of flavivirus envelope protein or premembrane and envelope protein domain three of DENV.

Due to the reports of a lower mortality rate among SARS-CoV-2 patients who previously experienced a DENV infection, this study investigated the possibility of cross-protection between arboviral antibodies and SARS-CoV-2 infection *in vitro*. Microneutralization assay revealed that there was some level of cross-neutralizing activity when tested against SARS-CoV-2 RVPs, a pseudotyped lentivirus containing SARS-CoV-2 spike-RBD, and a luciferase reporter gene. Antibodies acquired from febrile illness and DENV subpopulations could inhibit the RVPs from infecting HEK-293T-hsACE2. However, this activity disappeared when tested against the live SARS-CoV-2 virus. These data indicate an ability of DENV antibodies to bind to the spike 1-RBD, but this may not be enough to inhibit live virus infection. The results from this study indicate the possibility of antibody cross-reactivity when evaluated by a binding antibody assay but do not reflect cross-protection.

## MATERIALS AND METHODS

**Cell line, virus, and monoclonal antibodies.** HEK-293-human ACE2 expression cells (HEK-293T-hsACE2) were obtained from Integral Molecular, US. It is a permissive cell line used for infection by SARS-CoV-2 luciferase reporter virus particles (RVPs, Integral Molecular, US). Cells were grown in Dulbecco's minimum essential medium (DMEM, Invitrogen, US) supplemented with 10% heat-inactivated fetal bovine serum (HIFBS, Invitrogen, US), 10 mM HEPES (Invitrogen, US), 1% P&S, and 1.0 $\mu$g/mL puromycin, incubated at $35 \pm 2$°C and 5% $CO_2$. One-day-old cells were used for the measurement of neutralizing antibodies by SARS-CoV-2 luciferase RVP microneutralization assay. Vero E6 green monkey kidney epithelial cell lines were obtained from ATCC. Cells were grown in Eagle's minimum essential medium (EMEM, Invitrogen, US) supplemented with 5% heat-inactivated fetal bovine serum (HIFBS, Invitrogen, US), 1% L-glutamine, 1% P&S, 40 $\mu$g/mL gentamicin, and 0.25 $\mu$g/mL fungizone, incubated at $35 \pm 2$°C and 5% $CO_2$. One-day-old cells were used for the measurement of neutralizing antibodies by microneutralization assay.

A Wuhan linage SARS-CoV-2 virus (Hong Kong/VM20001061/2020, NR-52282) was obtained through BEI Resources (National Institute of Allergy and Infectious Diseases [NIAID], USA). Viruses were propagated in Vero E6 cells to generate sufficient titers (100 tissue culture infectious dose 50 [$TCID_{50}$]) for the microneutralization assay. All isolates were quantitated by $TCID_{50}$ using the Reed-Muench method based on eight replicates per titration (20).

Hybridoma cell lines of the flavivirus envelope protein (4G2), DENV premembrane protein (DENV group, 2H2), and DENV-1 through DENV-4 envelope protein domain III (1F1, 3H5, 8A1, and 1H10) were obtained through ATCC. Hybridoma cells were cultured in serum-free media to produce monoclonal antibodies and purified by HiTrap Protein G HP (GE Healthcare, Sweden). Monoclonal activity and specificity were determined before use in this study.

**RNA extraction.** To detect DENV or CHIKV genome in a serum sample, RNA was extracted from 140 $\mu$L of each sample using QIAamp Viral RNA Minikit (Qiagen, Valencia, USA). Extracted RNA was used later in DENV or CHIKV nested RT-PCR.

**DENV nested RT-PCR.** The DENV nested RT-PCR was performed following the protocol previously described (21). Two rounds of PCR were performed and included the first round of RT-PCR and the second round of nested PCR. The first round of RT-PCR was performed by using a pair of universal DENV forward and reverse primers, targeting the DENV capsid/premembrane genes region. The first round of RT-PCR was a single RT-PCR step, including an RT at 42°C for 60 min, immediately followed by PCR (35 cycles of thermocycling at 94°C for 30 sec, 55°C for 1 min, and 72°C for 2 min). The first round RT-PCR product (1:50 dilution) was used in the second round of nested PCR. This nested PCR was prepared by making a mixture of 5 primers, including a universal DENV forward primer (used in the first round RT-PCR) and 4 DENV serotype-specific reverse primers (each targeting a different DENV serotype). The samples were run for 25 cycles under the same PCR thermocycling conditions described in the first round PCR step. The nested PCR product was detected in agarose gel electrophoresis. PCR product samples underwent gel electrophoresis using a DENV genome-positive control. Specimens exhibiting DNA bands of 482, 119, 290, or 392 bp, respectively, were identified as DENV-1, DENV-2, DENV-3, or DENV-4 positive, respectively.

**CHIKV RT-PCR.** The modified nested RT-PCR method previously described (22) was used to detect the CHIKV genome. The first round of RT-PCR was performed by using outer primer pairs and included the RT step at 48°C for 45 min, held at 94°C for 2 min, and followed by the PCR step with 40 cycles of thermocycling at 94°C for 1 min, 60°C for 1 min, 68°C for 1 min, and followed by the hold at 68°C for 7 min. The first round RT-PCR product (1:50 dilution) was used in the second round nested PCR buffer with inner primer pairs. The nested PCR was performed by heat activation at 94°C for 2 min and followed by PCR step with 40 cycles of thermocycling at 94°C for 1 min, 60°C for 1 min, 72°C for 30 sec, and held at 72°C for 7 min. The nested PCR product was detected in agarose gel electrophoresis. A specimen containing CHIKV was identified by the detection of a DNA band of 200 bp in size on the gel loaded with nested PCR product corresponding to DNA amplified from the CHIKV genome in the positive control.

**Qualitative anti-SARS-CoV-2 IgA and IgG ELISAs.** Anti-SARS-CoV-2 spike 1 (S1) IgA and IgG ELISA (EUROIMMUN, Lübeck, Germany) were performed following the manufacturer's instructions. The ELISA utilized a 96-well plate coated with recombinant SARS-CoV-2 spike 1 protein, Wuhan strain. Positive, negative, and calibrator controls were used in each plate. The OD ratio of the controls and samples over the OD of the calibrator was calculated. Results were reported as negative (OD ratio <0.8), borderline (OD ratio 0.8 to <1.1), or positive (OD ratio $\geq$1.1).

Anti-SARS-CoV-2 IgA and IgG were also measured in the same set of acute and convalescent specimens using an AFRIMS in-house assay. Microtiter plates were coated with SARS-CoV-2 spike-RBD protein (Abcam, UK). Nonspecific binding proteins were blocked with 5% skim milk before the addition of serum samples. Goat anti-human IgA or IgG horseradish peroxidase (HRP) conjugate was added separately into each well for the detection of IgA or IgG, respectively. After adding the HRP substrate, the OD values were measured at 450 nM. Cutoff OD values for the positive result were identified as 2.9 and 3.2 times higher than the OD negative for IgA and IgG, respectively. The sensitivity of IgA and IgG in-house ELISA were 90.5% and 95.9%, respectively. The specificity was 97.1% for IgA and 96.3% for IgG in-house ELISA.

**Microneutralization assays (MNA).** There were two types of MNA utilized to measure the level of neutralizing (NT) antibodies against the SARS-CoV-2, Wuhan strain. The first MNA was a SARS-CoV-2 reporter virus particle (RVP) assay. This assay was performed in a BSL-2 laboratory using a pseudo reporter virus containing SARS-CoV-2 (Wuhan strain) spike-receptor binding domain (RBD), and a luciferase reporter gene (Integral Molecular, US). Before infection of susceptible cells expressing human ACE2 receptor (HEK-293T-hsACE2 cells), serum samples and positive/negative controls were heat-inactivated at 56°C for 30 min. Heat-inactivated samples and controls were diluted 2-fold in the culture medium and

then underwent serial 2-fold dilutions to a final concentration of 1:1280. Then, incubated with a fixed amount of RVP. The number of RVP-infected cells was determined by adding luciferase substrate and measuring luminescence with a luminometer. The ability to reduce the number of RVP-infected cells was compared to control cultures inoculated with RVP and diluent alone. The 50% neutralization titer ($NT_{50}$) was calculated using log probit analysis.

The second MNA used live SARS-CoV-2, Wuhan strain. All procedures were performed in a BSL-3 laboratory following a standard neutralization assay using a CPE-based colorimetric read-out (23–25). Cell (CC) and virus (VC) control wells were included. Before infecting Vero E6 cells in 96-well plates, serum samples and positive/negative controls were heat-inactivated at 56°C for 30 min. Heat-inactivated samples and controls were diluted 2-fold in the culture medium and then underwent serial 2-fold dilutions to a final concentration of 1:1280. Samples and controls were then incubated with 100 $TCID_{50}$ of SARS-CoV-2 virus at 37°C and 5% $CO_2$ for 1 h. The 96-well plates were then incubated at 37°C, 5% $CO_2$ for 5 days. Virus-infected cells were stained with 0.02% neutral red (Sigma, US) in $1\times$ PBS (Invitrogen, US). Lysis solution was added and incubated for 15 min at RT before measuring OD at 540 nm. The percentage of virus infectivity in VC and samples were calculated based on the OD of CC, infectivity (%) = (OD of CC − OD of the sample) $\times$ 100. The 50% neutralization titer ($NT_{50}$) was calculated using log probit analysis. Negative and positive controls were included.

**Statistical analysis.** Data analysis was performed using GraphPad Prism, version 9, and SPSS version 26. Probit analysis was used to determine the levels of 50% neutralizing antibody detected by RVP and live SARS-CoV-2 MN assay. The Mann-Whitney test was used to compare the level of SARS-CoV-2 IgA/IgG antibodies among all groups. $P < 0.05$ was considered statistically significant. Pearson correlation coefficient was used to determine the correlation between anti-SARS-CoV-2 spike 1 IgG with DENV IgM/IgG antibodies and the day after onset.

## ACKNOWLEDGMENTS

We acknowledged Bassam Hallis, Alex Sigal, and Tulio de Oliveira of BEI resources of the National Institute of Allergy and Infectious Diseases, National Institutes of Health, who provided the SARS-CoV-2, Wuhan strain lineage, isolate Hong Kong/VM20001061/2020, NR-52282.

This research study was supported by Armed Forces Health Surveillance Branch (AFHSB) and its Global Emerging Infectious Surveillance (GEIS) Section, United States, under grant number P0021_22_AF for the fiscal year 2022.

The material was reviewed by the Walter Reed Army Institute of Research. There was no objection to its presentation and/or publication. The opinions or assertions contained herein are the private views of the author and are not to be construed as official or as reflecting true views of the Department of the Army or the Department of Defense. The investigators have adhered to the policies for the protection of human subjects as prescribed in AR 70 to 25.

We declare no conflict of interest.

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
