## [Reviewer comments · Microbiology Spectrum]

Microbiology Spectrum

Anti-arbovirus antibodies cross-react to severe acute respiratory syndrome coronavirus 2

Taweewun Hunsawong, Darunee Buddhari, Kamonthip Rungrojcharoenkit, Rungarun Suthangkornkul, Duangrat Mongkolsirichaikul, Jindarat Lohachanakul, Kedsara Tayong, Kanittha Sirikajornpan, Prinyada Rodpradit, Yongyuth Poolpanichupatam, Chonticha Klungthong, Darunee Utennam, Surachai Kaewhiran, Thomas Cotrone, Stefan Fernandez, and Anthony Jones

Corresponding Author(s): Taweewun Hunsawong, US Army Medical Directorate of the Armed Forces Research Institute of Medical Sciences

Review Timeline:

Submission Date:	July 14, 2022
Editorial Decision:	September 11, 2022
Revision Received:	November 6, 2022
Accepted:	November 10, 2022

Editor: Leonidas Stamatatos

Reviewer(s): Disclosure of reviewer identity is with reference to reviewer comments included in decision letter(s). The following individuals involved in review of your submission have agreed to reveal their identity: Himadri Nath (Reviewer #1); Anton M Sholukh (Reviewer #2)

Transaction Report:

DOI: <https://doi.org/10.1128/spectrum.02639-22>

September 11, 2022

Dr. Taweewun Hunsawong
US Army Medical Directorate of the Armed Forces Research Institute of Medical Sciences
Virology
315/6 Ratchavithi Rd.,
Payathai, Ratchathevi
Bangkok, Bangkok 10400
Thailand

Re: Spectrum02639-22 (Anti-arbovirus antibodies cross-react to severe acute respiratory syndrome coronavirus 2)

Dear Dr. Taweewun Hunsawong:

The manuscript was reviewed by two experts on the field who raised several questions and are requesting specific clarifications. Please revise your manuscript accordingly.

Link Not Available

Sincerely,

Leonidas Stamatatos

Journals Department
Reviewer comments:

Reviewer #1 (Comments for the Author):

In this article, the authors have reported that antibodies against arbovirus (dengue, chikungunya) can cross-react with SARS-CoV-2. One of the possible outcomes of such cross-reactivity is neutralization which has also been tested using Reporter virus particle (expressing Spike-RBD) and SARS-CoV-2 Wuhan strain. While there are few published reports stating the cross-reactivity between dengue and SARS-CoV-2, the experiments to understand the possible neutralization of SARS-CoV-2 using anti-DENV Abs is the noble part of this study.

I would like to mention a few points which authors may consider-

1. In lines number 183-185, it has been mentioned that DENV and CHIKV samples were tested using RT-PCR. Authors may add the details in the methods section.
2. In the neutralization assay, the authors could have included the dilutions of SARS-CoV-2 Ab positive sera as a positive control to show the assay condition's robustness.
3. For neutralization assay using Wuhan SARS-CoV-2, authors initially screened the samples with SARS-CoV-2 RVP microneutralization assay. But as RVP contains only Spike-RBD glycoprotein, it has excluded the other cross-reacting serum samples (in ELISA). It could have been better if the authors could demonstrate/examine the neutralization potency of all the cross-reacting sera (in ELISA with Spike) using the SARS-CoV-2 Wuhan strain.
3. There is a report stating the neutralization of dengue virus using SARS-CoV-2 infected patients' sera i.e. the reverse scenario. This can be mentioned in this article to present the complete scenario for the readers.

Reviewer #2 (Comments for the Author):

Article by Hunsawong et al. "Anti-arbovirus antibodies cross-react to severe acute respiratory syndrome coronavirus 2"

Presented study attempts to test whether antibodies elicited to Dengue and Chikungunya infections cross-react with SARS-CoV-2 and thus can skew clinical testing. The question is important especially for regions with high rate of DENV and CHIKV infection. Despite importance for clinical diagnostics, the study has several major flaws. Size of pre-pandemic sample is small and does not allow proper cross-reactivity comparison. Study lacks direct evidence that observed cross-recognition of SARS-CoV-2 proteins by commercial and in-house assays is the result of presence of DENV or CHIKV-specific antibodies especially. Statistical methods are inappropriate, and interpretation is questionable. Method description lacks some important details.

Major points:

1. NHS sample size is very small and does not allow to properly establish positivity cutoff value. For that, at least 30 samples is needed. It is widely published in peer-reviewed articles that about 2-5% pre-pandemic samples show considerable readout values in binding assays against SARS-CoV-2 spike and about 0.5-1% yield high binding values for RBD. Therefore, positivity threshold established with only 6 samples can be substantially biased. Furthermore, low threshold results in calling more false positive results and thus skewing outcome of the study.
2. Because Euromune ELISA has built-in and validated positivity threshold it is reasonable to use it to evaluate positivity of the tested samples.
3. Line 128-137: What was the OD for negative IgA and IgG against that was developed a threshold of "2.9 and 3.2 times higher" Where these values are coming from?
4. What was the Reporter virus? Source? Was it prepared in-house or purchased?
5. Day 0-5 should be no IgG, day 5-16 is early convalescence and IgG level is usually low. Not clear what was the purpose of comparing acute vs convalescent samples? What new can we learn from that comparison?
6. Figure 1. Between what groups statistical significance was calculated with Mann-Whitney test? If this was multiple group comparison, then Mann-Whitney test can't be used. For that, Kruskal-Wallis test should be used if normal distribution is expected with Dunn's correction for multiple comparisons. If samples have normal distribution that can be actually tested, then One-way ANOVA can be applied but again with correction for multiple comparisons.
7. There is some correlation to IgM levels that are not presented as well as how was correlation performed. Cognate graphs are required as supplementary materials.
8. Correlation seemingly is not interpreted correctly. r values range between -1 to 1. All values close to 0 interpreted as no correlation. Overall r values below 0.5/-0.5 considered as weak association. P values for correlation must be provided.
9. Line 225: to investigate whether arboviral antibodies mediate cross-neutralization of SARS-CoV-2 to, one need to isolate monoclonal or affinity-purified SARS-CoV-2 specific antibodies from arbovirus seropositive persons and then test then in cross-neutralization assay. Neutralization by serum from arbovirus seropositive individuals cannot provide causative evidence as it is composed of many different antibody species and cross-binding does not define cross-neutralization that can come from antibodies targeting other, more related viruses such as OC43 and HKU1. This brings us to the point this study is trying to address - possible SARS-CoV-2 cross-recognition in DENV and CHIKV positive sera. To proof that cross-recognition of SARS-CoV-2 originates from prior infection with Dengue or Chikungunya virus, sera should be depleted of cognate antibodies using CHIKV or DENV antigens and then tested again in binding assay. Competition ELISA with pre-incubation with DENV and CHIKV antigens is also can be done instead of depletion. Without such direct-proof experiments, it is impossible to exclude that cross-reactivity is a result of presence of OC43 or HKU1 or other endemic coronavirus that has been demonstrated in multiple studies.
10. Line 256: "to investigate whether arboviral antibodies can mediate cross-neutralization of SARS-CoV-2" There is no direct testing of arborvirus antibodies in the assay. Thus, this question cannot be tested in such experiment. Direct use of arborvirus-specific antibodies or depletion should be used.
11. Authors should state in how many replicates experiments were performed and clearly describe how NT50 was calculated showing titration curves. Otherwise, it is not clear how NT50 equal 13 was derived if the first dilution was 1/10 and the next is 1/100 given 10-fold titration schema.

Staff Comments:

Preparing Revision Guidelines

Please return the manuscript within 60 days; if you cannot complete the modification within this time period, please contact me. If you do not wish to modify the manuscript and prefer to submit it to another journal, please notify me of your decision immediately so that the manuscript may be formally withdrawn from consideration by Microbiology Spectrum.

Reviewer comments:

Reviewer #1 (Comments for the Author):

In this article, the authors have reported that antibodies against arbovirus (dengue, chikungunya) can cross-react with SARS-CoV-2. One of the possible outcomes of such cross-reactivity is neutralization which has also been tested using Reporter virus particle (expressing Spike-RBD) and SARS-CoV-2 Wuhan strain. While there are few published reports stating the cross-reactivity between dengue and SARS-CoV-2, the experiments to understand the possible neutralization of SARS-CoV-2 using anti-DENV Abs is the noble part of this study.

I would like to mention a few points which authors may consider-

1. In lines number 183-185, it has been mentioned that DENV and CHIKV samples were tested using RT-PCR. Authors may add the details in the methods section.

Response: We thanks the reviewer for the suggestion, DENV and CHIKV RT-PCR methods have been added into method section, lines 244-279.

2. In the neutralization assay, the authors could have included the dilutions of SARS-CoV-2 Ab positive sera as a positive control to show the assay condition's robustness.

Response: The corrected dilution of sample, positive and negative controls were added into the method, lines 307-308 and lines 318-320 for RVP and live SARS-CoV-2 MN assays, respectively.

3. For neutralization assay using Wuhan SARS-CoV-2, authors initially screened the samples with SARS-CoV-2 RVP microneutralization assay. But as RVP contains only Spike-RBD glycoprotein, it has excluded the other cross-reacting serum samples (in ELISA). It could have been better if the authors could demonstrate/examine the neutralization potency of all the cross-reacting sera (in ELISA with Spike) using the SARS-CoV-2 Wuhan strain.

Response: Thank you the reviewer for mentioning this point. All spike ELISA positive samples have been tested by microneutralization assay using both RVP containing spike-RBD and the whole live SARS-CoV-2, Wuhan strain. To clarify this, the sentence "Anti-SARS-CoV-2 IgA (Fig. 2A, n=30) and IgG (Fig. 2B, n=12) positive samples by either EUROIMMUN or in-house ELISAs were tested for functional antibodies by both RVP and live SARS-CoV-2 microneutralization assays" has been added, lines 160-161.

4. There is a report stating the neutralization of dengue virus using SARS-CoV-2 infected patients' sera i.e. the reverse scenario. This can be mentioned in this article to present the complete scenario for the readers.

Response: We prefer to complete the scenario as the reviewer's recommendation. However, we conducted this study using the repository samples from arboviral study without new blood collection and since Thailand is the endemic area of DENV and CHIKV, we were unable to ensure that the SARS-CoV-2 infected patients are DENV and CHIKV naïve which might be interfere the assay results.

Reviewer #2 (Comments for the Author):

Article by Hunsawong et al. "Anti-arbovirus antibodies cross-react to severe acute respiratory syndrome coronavirus 2"

Presented study attempts to test whether antibodies elicited to Dengue and Chikungunya infections cross-react with SARS-CoV-2 and thus can skew clinical testing. The question is important especially for regions with high rate of DENV and CHIKV infection. Despite importance for clinical diagnostics, the study has several major flaws. Size of pre-pandemic sample is small and does not allow proper cross-reactivity comparison. Study lacks direct evidence that observed cross-recognition of SARS-CoV-2 proteins by commercial and in-house assays is the result of presence of DENV or CHIKV-specific antibodies especially. Statistical methods are inappropriate, and interpretation is questionable. Method description lacks some important details.

Major points:

1. NHS sample size is very small and does not allow to properly establish positivity cutoff value. For that, at least 30 samples is needed. It is widely published in peer-reviewed articles that about 2-5% pre-pandemic samples show considerable readout values in binding assays against SARS-CoV-2 spike and about 0.5-1% yield high binding values for RBD. Therefore, positivity threshold established with only 6 samples can be substantially biased. Furthermore, low threshold results in calling more false positive results and thus skewing outcome of the study.

Response: Many thanks the reviewer for comments and guidance. In this study, the NHS group was not used to establish positive cut-off for SARS-CoV-2 seropositivity among arboviral specimens since the positive cut-off was already validated before using in this study.

For the EUROIMMUN ELISA, we followed the manufacturer instruction.

Other assays including in-house anti-SARS-CoV-2 IgA/IgG ELISA and reporter virus particle (RVP) or live SARS-CoV-2 microneutralization assays have been validated using other set of known SARS-CoV-2 positive and negative samples. For in-house SARS-CoV-2 IgA/IgG ELISA, we used the ROC curve analysis to identify the positive cut-off and found that the OD values at equal or higher than 2.9 and 3.2 of negative control OD for IgA and IgG, respectively, displayed the assay performance up to 91.3% sensitivity/97.1% specificity and 94.3% sensitivity/96.3% specificity for SARS-CoV-2 IgA and IgG ELISA, respectively.

For microneutralization assay, 50% neutralization of virus infection in tested samples (Dilution at 1:2 to 1:1,280) comparing to virus control wells in each plate was used as positive cut-off and the 50% neutralization titers was calculated using log prohibit analysis. Kindly find references below,

1. Manenti A, Maggetti M, Casa E, Martinuzzi D, Torelli A, Trombetta CM, Marchi S, Montomoli E. 2020. Evaluation of SARS-CoV-2 neutralizing antibodies using a CPE-based colorimetric live virus micro-neutralization assay in human serum samples. *Journal of medical virology* 92:2096-2104.
2. Buathong R, Hunsawong T, Wacharapluesadee S, Guharat S, Jirapipatt R, Ninwattana S, Thippamom N, Jitsatja A, Jones AR, Rungrojchareonkit K, Lohachanakul J, Suthangkornkul R,

Tayong K, Klungthong C, Fernandez S, Putcharoen O. Homologous or Heterologous COVID-19 Booster Regimens Significantly Impact Sero-Neutralization of SARS-CoV-2 Virus and Its Variants. *Vaccines (Basel)*. 2022 Aug 15;10(8):1321. doi: 10.3390/vaccines10081321. PMID: 36016209; PMCID: PMC9415363.

3. Hunsawong T, Fernandez S, Buathong R, Khadthasrima N, Rungrojchareonkit K, Lohachanakul J, et al. Limited and Short-Lasting Virus Neutralizing Titers Induced by Inactivated SARS-CoV-2 Vaccine. *Emerg Infect Dis*. 2021;27(12):3178-3180. <https://doi.org/10.3201/eid2712.211772>

2. Because Euromune ELISA has built-in and validated positivity threshold it is reasonable to use it to evaluate positivity of the tested samples.

Response: For the EUROIMMUN ELISA, we followed the manufacturer instruction to evaluate seropositivity of the tested samples.

3. Line 128-137: What was the OD for negative IgA and IgG against that was developed a threshold of "2.9 and 3.2 times higher" Where these values are coming from?

Response: The negative control was added in triplicate wells of each plate and the acceptable OD values for negative control were less than 0.200. For the threshold of in-house SARS-CoV-2 IgA (2.9 times higher than OD negative control)/IgG (3.2 times higher than OD negative control) ELISA, we used the ROC curve analysis to identify the positive cut-off and found that the OD values at equal or higher than 2.9 and 3.2 of negative control OD for IgA and IgG, respectively, displayed the assay performance up to 91.3% sensitivity/97.1% specificity and 94.3% sensitivity/96.3% specificity for SARS-CoV-2 IgA and IgG ELISA, respectively.

4. What was the Reporter virus? Source? Was it prepared in-house or purchased?

Response: We have already mentioned about the reporter virus under microneutralization assay, lines 301-304 that "The SARS-CoV-2 Reporter Virus Particle (RVP) assay, which was performed in the BSL-2 laboratory using a pseudo-reporter virus containing SARS-CoV-2 (Wuhan strain) spike-receptor binding domain (RBD), and a luciferase reporter gene (Integral Molecular, US)".

5. Day 0-5 should be no IgG, day 5-16 is early convalescence and IgG level is usually low. Not clear what was the purpose of comparing acute vs convalescent samples? What new can we learn from that comparison?

Response: For DENV infection, the levels of IgG antibody starting observe on 5 and 2 days post onset for primary and secondary DENV infection, respectively (reference: St. John, A.L., Rathore, A.P.S. Adaptive immune responses to primary and secondary dengue virus infections. *Nat Rev Immunol* 19, 218–230 (2019). <https://doi.org/10.1038/s41577-019-0123-x>). For the purpose of measuring SARS-CoV-2 seropositivity in both acute and convalescent samples, we hypothesized that if the antibodies production during DENV, CHIKV or other febrile diseases infection has ability to cross-reactive with SARS-CoV-2, we suppose to find the cross-reactivity (IgA) since the early phase of symptom onset and this activity should maintain until convalescent phase (IgG).

6. Figure 1. Between what groups statistical significance was calculated with Mann-Whitney test? If this was multiple group comparison, then Mann-Whitney test can't be used. For that, Kruskal-Wallis test should be used if normal distribution is expected with Dunn's correction for multiple comparisons. If samples have normal distribution that can be actually tested, then One-way ANOVA can be applied but again with correction for multiple comparisons.

Response: For statistical analysis, we started with testing for normal distribution by Anderson-Darling test, D'Agostino & Pearson test, Shapiro-Wilk test and Kolmogorov-Smirnov test. We found that almost all groups except CHIKV group were not normal distribution so the unpaired non-parametric, Mann-Whitney test (without multiple comparison) was used to compare the mean between 2 groups of samples as following details, NHS vs primary DENV, NHS vs secondary DENV, NHS vs CHIKV, NHS vs Febrile illness, primary DENV vs secondary DENV, primary DENV vs CHIKV, primary DENV vs Febrile illness, secondary DENV vs CHIKV, secondary DENV vs Febrile illness and CHIKV vs Febrile illness.

7. There is some correlation to IgM levels that are not presented as well as how was correlation performed. Cognate graphs are required as supplementary materials.

Response: We have already mentioned that "Pearson correlation coefficient was used to determine the correlation between anti-SARS-CoV-2 spike 1 IgG with DENV IgM/IgG antibodies and day post onset." Under statistical analysis section, lines 334-336. Most of them showed negative and poor correlation only EUROIMMUN IgA and IgG showed moderate correlation with DENV IgM. The r values were mentioned under result section, SARS-CoV-2 IgA and IgG binding antibodies, lines 149-154.

8. Correlation seemingly is not interpreted correctly. r values range between -1 to 1. All values close to 0 interpreted as no correlation. Overall r values below 0.5/-0.5 considered as weak association. P values for correlation must be provided.

Response: We thanks the reviewer for the guidance, the r and p values for correlation were added under result section, SARS-CoV-2 IgA and IgG binding antibodies, lines 149-154.

9. Line 225: to investigate whether arboviral antibodies mediate cross-neutralization of SARS-CoV-2 to, one need to isolate monoclonal or affinity-purified SARS-CoV-2 specific antibodies from arbovirus seropositive persons and then test then in cross-neutralization assay. Neutralization by serum from arbovirus seropositive individuals cannot provide causative evidence as it is composed of many different antibody species and cross-binding does not define cross-neutralization that can come from antibodies targeting other, more related viruses such as OC43 and HKU1. This brings us to the point this study is trying to address - possible SARS-CoV-2 cross-recognition in DENV and CHIKV positive sera. To proof that cross-recognition of SARS-CoV-2 originates from prior infection with Dengue or Chikungunya virus, sera should be depleted of cognate antibodies using CHIKV or DENV antigens and then tested again in binding assay. Competition ELISA with pre-incubation with DENV and CHIKV antigens is also can be done instead of depletion. Without such direct-proof experiments, it is impossible to exclude that cross-reactivity is a result of presence of OC43 or HKU1 or other endemic coronavirus that has been demonstrated in multiple studies.

Response: Many thanks to the reviewer for your suggestions. In this study, we used the repository samples under arboviral studies without new blood collection. The samples volume was quite limited and most of them were not enough to do the depletion of cognate antibody and competitive ELISA. However, we acknowledged that your suggestion was very valuable and will try to incorporate those assays in the future study.

10. Line 256: "to investigate whether arboviral antibodies can mediate cross-neutralization of SARS-CoV-2" There is no direct testing of arborvirus antibodies in the assay. Thus, this question cannot be tested in such experiment. Direct use of arborvirus-specific antibodies or depletion should be used.

Response: We changed the word "arboviral antibodies" to "antibodies production during arboviral infection".

11. Authors should state in how many replicates experiments were performed and clearly describe how NT50 was calculated showing titration curves. Otherwise, it is not clear how NT50 equal 13 was derived if the first dilution was 1/10 and the next is 1/100 given 10-fold titration schema.

Response: The three independent experiments were done. We already mentioned how the NT50 was calculated under method section, microneutralization assay and also corrected the dilution of samples including negative and positive controls from 1:10 to 1:2 serial dilution starting from 1:2 to 1:1,280, lines 307-308 and lines 318-320 for RVP and live SARS-CoV-2 MN assays, respectively.

November 10, 2022

Dr. Taweewun Hunsawong
US Army Medical Directorate of the Armed Forces Research Institute of Medical Sciences
Virology
315/6 Ratchavithi Rd.,
Payathai, Ratchathevi
Bangkok, Bangkok 10400
Thailand

Re: Spectrum02639-22R1 (Anti-arbovirus antibodies cross-react to severe acute respiratory syndrome coronavirus 2)

Dear Dr. Taweewun Hunsawong:

Your manuscript has been accepted, and I am forwarding it to the ASM Journals Department for publication. You will be notified when your proofs are ready to be viewed.

Sincerely,

Leonidas Stamatatos
Editor, Microbiology Spectrum
